# Association between Carbapenem Consumption and Clinical Outcomes in an In-Hospital Setting: Analysis of a Japanese Nationwide Administrative Database in 2020

**DOI:** 10.3390/antibiotics11121807

**Published:** 2022-12-13

**Authors:** Kozue Yamaguchi, Masayuki Maeda, Norio Ohmagari, Yuichi Muraki

**Affiliations:** 1Division of Infection Control Sciences, Department of Clinical Pharmacy, School of Pharmacy, Showa University, Tokyo 142-8555, Japan; 2AMR Clinical Reference Center, Disease Control and Prevention Center, National Center for Global Health and Medicine, Tokyo 162-8655, Japan; 3Department of Clinical Pharmacoepidemiology, Kyoto Pharmaceutical University, Kyoto 607-8414, Japan

**Keywords:** antimicrobial consumption, defined daily dose, diagnosis procedure combination data analysis, carbapenem consumption clinical outcomes

## Abstract

This study aimed to clarify the relationship between carbapenem consumption and clinical outcome using the diagnosis procedure combination (DPC) payment system database (2020) published by the Ministry of Health, Labour, and Welfare of Japan. This study divided 5316 medical facilities subject to aggregation into five facilities and calculated the median values, including facility characteristics, clinical outcomes, and carbapenem consumption. Next, a correlation analysis was performed between carbapenem consumption and clinical outcome, as well as a multiple regression analysis between carbapenem consumption as the dependent variable and clinical outcome, bed size, and proportion of patients by disease as independent variables. Additionally, three clinical outcomes available from the DPC payment system database were selected, including cure, readmission within 4 weeks, and the average length of stay. This study revealed no relationship between carbapenem consumption and clinical outcome in university hospitals and university hospital-equivalent community hospitals; however, a relationship was suggested in the community, DPC-prepared, and non-DPC hospitals. University hospitals and university hospital-equivalent community hospitals with a high consumption of carbapenems may need to reconsider the classification because of the limited number of facilities in this classification.

## 1. Introduction

Carbapenems, which are broad-spectrum antibacterial agents, have a high degree of tissue penetration and antibacterial activity against most Gram-negative bacilli, including antimicrobial-resistant (AMR) *Enterobacterales* and glucose-nonfermenting bacteria [1]. Guidelines recommend carbapenems as the first-line drug for severe infections due to resistant bacteria such as extended-spectrum beta-lactamase (ESBL)-producing bacteria [2]. The overuse of carbapenems, which have high efficacy and safety, is a concern because it is selected as an empirical treatment before identifying the causative bacterium and infected organ [1]. Therefore, carbapenem-resistant *Enterobacterales* (CRE) has become a major problem worldwide [3,4]. Additionally, the number of CRE detection has increased in Japan over the years [5,6]. The Japanese government launched a National Action Plan for AMR, and efforts are made at each facility to reduce unnecessary antimicrobial consumption, including carbapenems [7].

Carbapenem consumption should be kept to a minimum to conserve carbapenem efficacy in the future [8,9,10,11]. In Japan, carbapenem consumption accounted for approximately 10% of all parenteral antimicrobial consumption [12]. A previous study revealed that most carbapenems consumption included meropenem and doripenem in Japan because ertapenem was not approved [13]. Terahara et al. revealed a positive correlation between carbapenem consumption and carbapenem-resistant *P. aeruginosa* isolation in Japan [14]. Therefore, an antibiotic restriction, which is one of the antimicrobial stewardship programs, has been recommended to reduce unnecessary antibiotic consumption [15]. The Japanese medical reimbursement system for antimicrobial stewardship required restriction of in-hospital broad-spectrum antibiotic consumption such as carbapenems [16]. An antibiotic registration or notification system was implemented in Japanese acute hospitals to restrict inappropriate carbapenem consumption by the antimicrobial stewardship team (AST) [17,18]. However, patients’ poor clinical outcomes due to excessive carbapenem restriction should be avoided.

The diagnosis procedure combination (DPC) payment system is a comprehensive assessment system based on diagnostic group classification for acute hospitalization [19,20]. DPC was developed as a measuring tool for transparent inpatient care to standardize Japanese medical care, as well as evaluate and improve its quality. Medical institution categories have been classified as the university hospital group, university-equivalent community hospital group, and community hospital group. Over 60% of hospitals in Japan submit “DPC data” to the Ministry of Health, Labor, and Welfare (MHLW). DPC data include the discharge abstract and administrative claims data of inpatients. The MHLW collects the data for the purpose of health policy planning, including the DPC-based reimbursement system. The MHLW disclosed DPC data, including carbapenem-defined daily dose (DDD) in each hospital [21]. These data are suitable for analysis of the association between carbapenem consumption and various medical data in hospital settings. However, existing antibiotic consumption metrics, including DDD, and days of therapy do not reflect both antimicrobial appropriateness and indications [22,23]. Monitoring of conventional antimicrobial metrics alone is insufficient to evaluate patient outcomes [23,24,25]. Specific data on an association between carbapenem consumption and the patient’s clinical outcomes remains lacking [17,18]. Thus, the present study aimed to reveal the association between hospital carbapenem consumption and clinical outcomes, including discharge, readmission, and length of stay using a national database.

## 2. Results

### 2.1. Characteristics of Hospitals Stratified by Classification of Payment System

This study analyzed the DPC payment database on carbapenem consumption (presented as DDDs per 1000 patient days) that comprise 5316 healthcare facilities collected by the MHLW in the fiscal year 2020. This study included 82 university hospitals, 156 university hospital-equivalent community hospitals, 1510 community hospitals, 242 DPC-prepared hospitals, and 2918 non-DPC (fee-for-service payment) hospitals after excluding 408 facilities with missing data (Figure 1). Table 1 shows the characteristics and clinical outcomes of each facility stratified by facility type. University hospitals and university hospital-equivalent community hospitals consisted of large-sized hospitals, and community and non-DPC hospitals consisted of small to medium hospitals.

Figure 2 shows the carbapenem consumption stratified by facility type. As a result of the Kruskal-Wallis test, significant differences in carbapenem consumption were observed among the five groups (*p* < 0.001). There was no significant difference only between the university hospitals and the university-equivalent community hospitals. The median DDDs/1000 patient days in university hospitals were higher compared with community hospitals and non-DPC hospitals. Carbapenem consumption in community hospitals and non-DPC hospitals had a large dispersion.

### 2.2. Correlation Analysis between Carbapenem Consumption and Clinical Outcomes Stratified by Facility Type

Table 2 shows the correlation analysis between carbapenem consumption and clinical outcome for each facility type. No strong correlation was found between DDDs/patient days and each clinical outcome in each facility type (all rho < 0.5).

### 2.3. Factors for Carbapenem Consumption Using a Linear Regression Model

Table 3 shows the results of a linear regression model of factors for hospital carbapenem consumption. Only MDCs were associated with carbapenem consumption in university hospitals (MDC 3 and 6) and university hospital-equivalent community hospitals (MDC 13). In community hospitals and non-DPC hospitals, various MDCs (e.g., MDC 4 and 13), and clinical outcomes including discharge with cure, readmission rate, and length of stay were associated with carbapenem consumption.

## 3. Discussion

To our best knowledge, this is the first study to reveal the association between carbapenem consumption and major clinical outcomes in various facility types based on DPC database analysis. The large dispersion of carbapenem consumption in community and non-DPC hospitals would indicate concern about inappropriate carbapenem use in these hospitals.

In Japan, large-sized hospitals, such as university hospitals and university hospital-equivalent community hospitals, mainly hospitalize acute and critical illness cases, and transfer them to small or medium-sized hospitals after treatment. Conversely, patients admitted at small- and medium-sized hospitals are transferred to large hospitals when their disease becomes severe. Each prefecture in Japan is classified by medical region and these prefectures provide a medical care system in cooperation [26]. This study revealed no association between carbapenem consumption and clinical outcomes, despite a high median carbapenem consumption in large-sized hospitals compared with the community hospitals, DPC-prepared hospitals, and non-DPC hospitals. AST in large-sized hospitals that have sufficient human resources could facilitate appropriate carbapenem use according to their various inpatient backgrounds [9,27,28,29]. Whereas, carbapenem consumption was associated with the increment of discharge with cure and readmission in community, and non-DPC hospitals. Several studies and guidelines have supported the clinical efficacy of carbapenems [2,30,31]. However, the increment in carbapenem consumption was also associated with the readmission rate. These inconsistent results may be caused by different disease severity and patient comorbidities. The increment of readmission would be caused by disease severity and complicated comorbidities other than infectious diseases, although carbapenems are highly efficacious and could improve patient outcomes. The MDCs and clinical outcomes were extracted as associated factors for carbapenem consumption in community and non-DPC hospitals because patients admitted at small- and medium-sized hospitals, including those that transferred from large-sized hospitals, have various backgrounds.

Our previous DPC database analysis revealed that various MDCs were significantly associated with an increment in carbapenem consumption [32]. The stratification result by facility type in this study revealed a few associated factors in university hospitals and university hospital-equivalent community hospitals. Detecting the specific associated factors in carbapenem consumption would be difficult, considering the similar medical functions and inpatient backgrounds in these hospitals. Meanwhile, respiratory diseases (MDC4) and hematologic diseases (MDC13) were significantly associated with the increment of carbapenem consumption in community hospitals consistent with the previous study [32]. Rhodes et al. revealed the predictors of carbapenem consumption across North American hospitals [33]. The linear mixed-effects model revealed that non-carbapenem consumption, antibiogram publication, and license bed size were associated with carbapenem consumption [33]. Additionally, the rates of carbapenem consumption differed greatly among the hospitals. Our study revealed that inpatient backgrounds affect carbapenem consumption, which reinforces the importance of carbapenem consumption benchmarks according to each hospital’s characteristics.

The burden of carbapenem overuse associated with antibiotic selective pressure would be alarming nationwide, considering the number of community and non-DPC hospitals [12,14]. Various inpatient backgrounds and clinical outcomes in community hospitals were associated with carbapenem consumption. Additionally, the heterogeneity of carbapenem consumption was speculated. Carbapenem overuse and misusage in these hospitals would be concerning. A previous study recommended that institutional guidelines for carbapenem use should focus on four common infectious diseases, such as respiratory, genitourinary, intra-abdominal, and bloodstream, to promote carbapenem stewardship [34]. However, previous investigations revealed that small- to medium-size hospitals had limited resources to promote antimicrobial stewardship [18,35,36]. Increased commitment by policymakers and stakeholders to support community and non-DPC hospitals would be needed using the benchmark of the carbapenem consumption metric.

Several important study limitations should be noted. First, the patient individual background (e.g., comorbidities) and severity indices (e.g., sequential organ failure assessment score) were not available from the DPC database. Moreover, the population of this study included patients with diseases other than infectious diseases. An available database of specific infectious diseases and antimicrobial consumption, including patient and facility data, has been limited in Japan. Second, a metric for antimicrobial consumption other than carbapenems was not analyzed because the DPC database has not disclosed antimicrobial consumption other than carbapenems. Metrics for each antimicrobial categorization akin to the standardized antimicrobial administration ratio can be developed, provided that the MHLW discloses various antimicrobial consumption [37]. Standardized metrics combined with patients’ clinical outcomes will be utilized [38]. The previous multicenter study in Japan reported the correlation between carbapenem use and the rate of carbapenem-resistant Pseudomonas aeruginosa using the Japanese surveillance system [39]. The development of a database for infectious diseases and antimicrobial consumption that is accessible from every Japanese hospital would be needed to evaluate the appropriate antibiotic consumption in each hospital [39,40].

## 4. Materials and Methods

### 4.1. Data Sources

This study analyzed data on carbapenem consumption (presented as DDDs per 1000 patient days) of inpatients across 5316 healthcare facilities collected by the MHLW in the fiscal year 2020 [21]. Imipenem, meropenem, doripenem, panipenem, and biapenem injections were approved for consumption in Japan.

The data included clinical outcomes at discharge (rate of cure, remission, stable disease, exacerbation, death, and others), readmission and length of stay, number of beds, and MDCs. MDCs were principal diagnoses (from the International Classification of Diseases 10th Revision (ICD-10)) of inpatients into the following 18 mutually exclusive diagnosis areas: diseases and disorders of the (1) nervous system; (2) eye; (3) ears, nose, mouth, and throat; (4) respiratory system; (5) circulatory system; (6) digestive system, hepatobiliary system, and pancreas; (7) musculoskeletal system; (8) skin and subcutaneous tissue; (9) breast; (10) endocrine, nutritional, and metabolic system; (11) kidney, urinary tract, and male reproductive system; (12) female reproductive system, pregnancy, childbirth, and puerperium; (13) blood and blood-forming organs, and immunological disorders; (14) newborn and other neonates with conditions originating from the perinatal period; (15) pediatric; (16) injuries, burns, poisoning, and the toxic effect of drugs; (17) mental; and (18) others. MDCs were a group of 18 diseases based on the ICD-10, and evaluating the differences in the background of inpatients at each medical institution as a factor is possible using this ratio.

### 4.2. Statistical Analysis

Hospital characteristics of each facility type were presented as median and interquartile range owing to the non-normal distribution. Carbapenem DDDs/1000 patient days in each facility type were summarized using a boxplot. Multiple comparisons were performed using the Kruskal-Wallis test with Bonferroni correction. Spearman’s rank correlation coefficient was utilized to investigate the possible association between the carbapenem consumption and the major clinical outcomes (discharge with cure, readmission within 4 weeks, and average length of stay) in each facility type.

Then, we performed a multivariable linear regression analysis using a stepwise method for each facility type to identify factors related to carbapenem consumption. We selected clinical outcomes (discharge with cure, readmission within 4 weeks, and average length of stay), the number of hospital beds (per 100 beds increment), and each of the MDCs as covariates.

Multicollinearity was assessed using the Spearman’s rank correlation coefficient and variance inflation factor. All the statistical analyses were two-tailed, with *p*-values of <0.05 indicating statistical significance. Statistical analyses were performed using the Statistical Package for the Social Sciences version 27.0 (IBM Japan, Tokyo, Japan).

This study required no ethical approval or informed consent because only publicly accessible data on the MHLW homepages were obtained.

## 5. Conclusions

In conclusion, carbapenem consumption in community and non-DPC hospitals was associated with major clinical outcomes and inpatient characteristics based on national database analysis. Both carbapenem overuse and underuse are alarming for patient clinical outcomes and antimicrobial resistance. Thus, we revealed the model of carbapenem consumption, including patients and hospital characteristics. Further studies are needed to develop uniformly applied metrics to evaluate appropriate antimicrobial consumption across diverse healthcare settings.

## Figures and Tables

**Figure 1 antibiotics-11-01807-f001:**
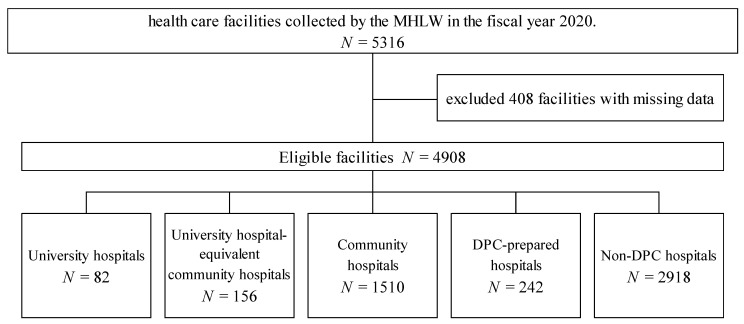
Study flow diagram showing the stratification and selection of patients in the Diagnosis Procedure Combination database by the Ministry of Health and Welfare in the fiscal year 2020.

**Figure 2 antibiotics-11-01807-f002:**
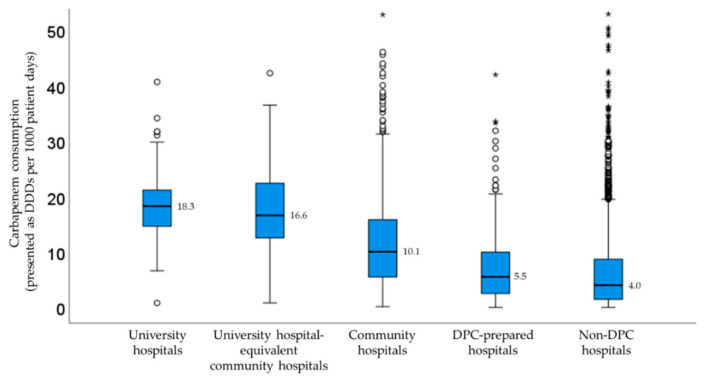
Median carbapenem consumption stratified by facility type. The outliners were plotted on the box plot as a small circle and a star.

**Table 1 antibiotics-11-01807-t001:** Characteristics of hospitals stratified by facility type.

	University Hospitals (*N* = 82)	University Hospital-Equivalent Community Hospitals (*N* = 156)	Community Hospitals (*N* = 1510)	DPC-Prepared Hospitals (*N* = 242)	Non-DPC Hospitals (*N* = 2918)
Number of licensed beds	821 (643–968)	578 (477–675)	260 (180–367)	148 (102–199)	107 (64–168)
Outcome at discharge					
Cure (%)	77.17 (71.97–83.85)	81.24 (77.09–85.71)	82.66 (76.84–87.04)	81.80 (72.69–87.62)	76.83 (62.61–86.21)
Remission (%)	0.23 (0.066–0.406)	0.13 (0.03–0.56)	0.06 (0–0.31)	0.03 (0–0.29)	0 (0–0.66)
Stable disease (%)	12.54 (5.68–18.03)	8.08 (3.58–11.60)	6.15 (2.79–10.89)	6.37 (2.67–12.28)	7.05 (2.45–13.55)
Exacerbation (%)	0.063 (0.024–0.13)	0.03 (0.01–0.13)	0.12 (0.03–0.32)	0.40 (0.11–0.84)	0.63 (0–1.76)
Death (%)	1.43 (1.08–1.795)	2.57 (2.04–3.19)	3.62 (2.51–5.12)	4.41 (2.56–6.75)	6.87 (2.93–12.22)
Others (%)	8.54 (5.68–10.54)	7.11 (5.44–8.98)	5.22 (2.74–7.96)	2.27 (0.43–6.22)	0.65 (0–4.37)
Readmission within 4 weeks (%)	14.51 (12.68–16.55)	12.47 (10.55–14.69)	9.93 (7.31–12.76)	8.04 (4.50–12.01)	5.73 (3.14–9.75)
Average length of stay (day)	12.14 (11.60–12.98)	11.39 (10.81–12.14)	12.15 (10.71–13.54)	12.78 (10.49–15.30)	14.46 (10.93–18.22)

Data were presented as median (interquartile range).

**Table 2 antibiotics-11-01807-t002:** Correlation analysis results of carbapenem consumption stratified by facility type, cure, readmission within 4 weeks, and average length of stay.

	Spearman’s Correlation Coefficient (ρ)	*p*-Value
University hospitals		
Cure rate	−0.068	0.541
Rate of readmission within 4 weeks	−0.311	0.004
Average length of stay	−0.001	0.993
University hospital-equivalent community hospitals		
Cure rate	−0.111	0.167
Rate of readmission within 4 weeks	0.13	0.098
Average length of stay	0.021	0.792
Community hospitals		
Cure rate	−0.056	0.03
Rate of readmission within 4 weeks	0.342	<0.001
Average length of stay	0.007	0.794
DPC-prepared hospitals		
Cure rate	−0.113	0.081
Rate of readmission within 4 weeks	0.483	<0.001
Average length of stay	−0.014	0.829
Non-DPC hospitals		
Cure rate	−0.098	<0.001
Rate of readmission within 4 weeks	0.265	<0.001
Average length of stay	0.005	0.811

DPC, diagnosis procedure combination.

**Table 3 antibiotics-11-01807-t003:** Multivariable linear regression models of associated factors of carbapenem consumption by facility type.

Factors	Partial Regression Coefficient (95% Confidence Interval)		*p*-Value
Model 1 ^a^ (university hospitals)			
MDC 03 (ear, nose, mouth, and throat)	106.643	(5.581–207.705)	0.039
MDC 16 (injuries, burns, poisoning, and toxic effect of drugs)	133.498	(22.872–244.124)	0.019
Constant term	9.683	(3.800–15.565)	0.002
Model 2 ^b^ (university hospital-equivalent community hospitals)			
MDC 13 (blood and immunological disorders)	92.997	(32.170–153.824)	0.003
Constant term	13.930	(11.352–16.508)	0.000
Model 3 ^c^ (community hospitals)			
Cure rate	9.955	(5.687–14.222)	0.000
Rate of readmission within 4 weeks	15.361	(5.878–24.843)	0.002
Average length of stay	0.176	(0.016–0.335)	0.031
Licensed bed size *	1.188	(0.931–1.445)	0.000
MDC 01 (nervous system)	−4.911	(−8.368 to −1.454)	0.005
MDC 04 (respiratory system)	8.127	(2.221–14.034)	0.007
MDC 07 (musculoskeletal system)	−12.784	(−17.522 to −8.046)	0.000
MDC 09 (breast)	−16.717	(−27.749 to −5.685)	0.003
MDC 10 (endocrine, nutritional, and metabolic system)	−19.321	(−31.693 to −6.950)	0.002
MDC 13 (blood and immunological disorders)	62.499	(48.049–76.950)	0.000
MDC 14 (newborn)	−24.892	(−34.774 to −15.009)	0.000
MDC 16 (injuries, burns, poisoning, and toxic effect of drugs)	−7.841	(−13.039 to −2.642)	0.003
MDC 17 (mental disorders)	−169.432	(−256.636 to −82.227)	0.000
Model 4 ^d^ (DPC-prepared hospitals)			
Cure rate	12.344	(4.946–19.742)	0.001
Rate of readmission within 4 weeks	31.157	(14.330–47.984)	0.000
MDC 02 (eye)	−10.099	(−19.786 to −0.413)	0.041
MDC 04 (respiratory system)	24.198	(14.422–33.975)	0.000
MDC 11 (kidney, urinary tract, and male reproductive system)	13.260	(5.408–21.112)	0.001
MDC 15 (pediatric)	−178.586	(−302.983 to−54.190)	0.005
MDC 16 (injuries, burns, poisoning, and toxic effect of drugs)	−9.409	(−16.880 to −1.938)	0.014
Model 5 ^e^ (non-DPC hospitals)			
Cure rate	1.890	(0.283–3.498)	0.021
Rate of readmission within 4 weeks	19.484	(14.966–24.002)	0.000
MDC 04 (respiratory system)	16.127	(13.091–19.162)	0.000
MDC 05 (circulatory system)	4.174	(1.417–6.931)	0.003
MDC 06 (digestive and hepatobiliary system, and pancreas)	4.790	(3.475–6.104)	0.000
MDC 11 (kidney, urinary tract, and male reproductive system)	7.595	(5.144–10.047)	0.000
MDC 13 (blood and immunological disorders)	26.154	(17.447–34.861)	0.000

DPC, diagnosis procedure combination; MDC, major diagnostic. MDC No. (diagnosis areas). * Indicates increments of 100 beds. Stepwise method: ^a^ R^2^ = 0.109, Durbin-Watson = 1.690; ^b^ R^2^ = 0.056, Durbin-Watson = 1.993; ^c^ R^2^ = 0.250, Durbin-Watson = 1.892; ^d^ R^2^ = 0.265, Durbin-Watson = 2.098; ^e^ R^2^ = 0.130, Durbin-Watson = 1.980.

## Data Availability

Publicly available datasets were analyzed in this study. These data can be found here: [https://www.mhlw.go.jp/stf/shingi2/0000196043_00005.html (accessed on 20 April 2022)].

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
