# Peer review of "Association between Carbapenem Consumption and Clinical Outcomes in an In-Hospital Setting: Analysis of a Japanese Nationwide Administrative Database in 2020"

_antibiotics, 2022, doi:10.3390/antibiotics11121807_

Round 1
Reviewer 1 Report
This is an interesting study focusing on the consumption of carbapanems and the clinical outcome in various clinical settings in Japan. The paper is well written and the results are interesting and well founded. Carbapenems are an important group of broad-spectrum antibiotics which should be reserved for strains producing b-lactams such as ESBL isolates. Consumption of carbapenems has been directly associated with the appearance and increase of prevalence of isolates producing carbapanemases (KPC, MBL, OXA etc). It would be also important if the authors could provide any clues about the prevalence of such isolates in Japan in the settings described in their study and associate those data with their findings. Also a comment including any studies performed in Japan on the epidemiology of such isolates would also add important information in their conclusions.
Author Response
Reviewer #1:
Thank you very much for reviewing our manuscript together with the comments. The suggestions for improving the manuscript were very helpful in preparing a revised submission. We revised the manuscript according to your suggestions.
This is an interesting study focusing on the consumption of carbapanems and the clinical outcome in various clinical settings in Japan. The paper is well written and the results are interesting and well founded. Carbapenems are an important group of broad-spectrum antibiotics which should be reserved for strains producing b-lactams such as ESBL isolates. Consumption of carbapenems has been directly associated with the appearance and increase of prevalence of isolates producing carbapanemases (KPC, MBL, OXA etc). It would be also important if the authors could provide any clues about the prevalence of such isolates in Japan in the settings described in their study and associate those data with their findings. Also a comment including any studies performed in Japan on the epidemiology of such isolates would also add important information in their conclusions.
Response:
Thank you for the comment. We added the information.
Change: P7 L195 – 197
The previous multicenter study in Japan reported the correlation between carbapenem use and the rate of carbapenem-resistant Pseudomonas aeruginosa using the Japanese surveillance system [39].

Reviewer 2 Report
I attached a word with my comments

Author Response
Reviewer #2:
Thank you very much for reviewing our manuscript together with the comments. The suggestions for improving the manuscript were very helpful in preparing a revised submission. We revised the manuscript according to your suggestions.
1. L84-85 “Figure 2 shows the carbapenem consumption stratified by facility type. The median 84 DDDs/1000 patient days in university hospitals were higher compared with community 85 hospitals and non-DPC hospitals” Please add the total p-value (Kruskall-Wallis one-way analysis of variance). Then, I highly recommend to also run a non-parametric post hoc analysis with Bonferroni correction for all different pairs of hospitals in order to identify exactly which groups differ from each other (this will result to a number of 10 different sub-analyses and p-values).
Response:
Thank you for the comment. According to the comment, we have added statistical analysis to results and methods.
Change: P2 L90 ‐ 93 and P8 L224 ‐ 226
2. In L108-109 “No strong correlation was found between DDDs/patient days and each clinical outcome in each facility type”. From Figure 3, I observe a significant and positive association between DDDs/patient days and re-admission within 4 weeks (p<0.001) in Community, DPC-prepared and Non-DPC prepared hospitals.
Response:
As pointed out by the reviewer, a statistically significant correlation was observed. However, the correlation coefficient was small (<0.5), so no strong correlations were observed in this correlation analysis. Typically, we think that over 0.7 of the coefficient was interpreted strong correlation.
3. In Figure 3 (L123) you refer as “Spearman’s correlation analysis results”, but in the methods you report Pearson’s rank correlation, please clarify. I agree using Spearman’s rho instead of Pearson’s r in these data, since they are not normally distributed, but if this is the case you should change your sentence in L229-231 to “Spearman’s rho correlation coefficient was utilized to investigate possible association between…”
Response:
Thank you for your advice. We reworked with Spearman's rank correlation coefficient.
Change: Table 2 and P8 L225 - 226 and P8 L236
4. L129-131 “In community hospitals and non-DPC hospitals, various MDCs and clinical outcomes including discharge with cure, readmission rate, and length of stay were associated with carbapenem consumption.” Please describe what was the association with numbers, don’t include all of them, but some that have clinical significance
Response:
We have stated specific factors with statistical and clinical significance.
Change: P5 L122 ‐ 123
5. In Table 2 (L133-134) if I understand correct you performed one multivariable linear regression model for each group of hospitals separately and thus i.e. for the model with university hospitals the sample size is equal to N=82? If yes, you should refer in the title as “Multivariable linear regression models” instead of “Final linear regression model”.
In addition, it is not clear to me why did you chose these covariates in each model. Why do the covariates differ between each model (i.e. in community hospitals you have included the covariate “MDC07 (musculoskeletal system)”, but it is not included in the models with the rest hospitals? Did you use any specific statistical and/or clinical method (i.e. stepwise backward, forward, with AIC)? I also recommend adding first a Table with a full multivariable linear regression model, including all hospitals (N=4,908) (each group with a separate dummy variable and a group as reference category) and then procced with Table 2.
Response:
Thank you for the comments. We revised the description of “Final linear regression model” to “Multivariable linear regression models”
We selected the stepwise method, so we showed only significant factors at the final model.
We included all variables stated in the Methods section. Considering the busyness of the table due to lots of variables (>20), we did not include the factors which were not statistical significance in the Table.
In addition, we reported the preliminary results of regression analysis including all hospitals previously (ref. 32). So, we focused on the stratification by facility type in this study.
Change: P5 L125, Footnote to Table 3 and P8 L230 ‐ 231
6. L151-152 “Whereas, carbapenem consumption was associated with the increment of discharge with cure and readmission in community and non-DPC hospitals.” Cure and readmission were associated also with DPC-prepared hospitals, based on Table 2.
Response:
Thank you for the comment. We have revised it.
Change: P6 L142
7. L233 “linear multiple regression analysis” change to “multivariable linear regression analysis”
Response:
Thank you for the comment. We have revised it.
Change: P8 L230
8. L233-234 “Then, we performed linear multiple regression analysis for each facility type to identify factors related to carbapenem consumption.” Did you also check if the model fitted well? (i.e. homoskedasticity, normality of residuals, outliers). If yes, you should mention it in the statistical analysis section in methods
Response:
We assessed several factors relevant to the fitness of model. We have added the results to the footnote of the Table 3.
Change: P6 Footnote to Table 3
9. L235 “(discharge with cure, discharge with cure, readmission within 4 weeks, and average length of stay)” I don’t understand why you mention “discharge with cure” twice, is it by mistake?
Response:
Thank you for pointing that out. We apologize for the mistake. We corrected the duplicated "discharge with cure".
Change: P8 L232
10. L236-237 “…the proportion of long-term care beds, the proportion of psychiatric beds, and each MDCs as covariates.” How long-term care beds were defined, what cut-off was used? Also why did you specially choose psychiatric beds as a covariate, please clarify.
Response:
Thank you for the comment. Following to comment of reviewer #3, we excluded the factors of long-term care beds and psychiatric beds. Although we collected the data from the DPC database, these factors are not considered relevant to carbapenem consumption typically. Moreover, almost all of the hospitals in this study do have not these beds as shown in Table. We performed a regression analysis again and the results were identical. The descriptions related to these have been revised.
Change: Table 1 and P8 L234 - 235
11. L227-228 “Hospital characteristics of each facility type were presented as median and interquartile range.” Did you check for normality and chose median? How? You should mention that
Response:
Thank you for the comment. We agree with the check for normality of continuous variables. However, we considered that variables of hospital characteristics such as the number of beds were close to the ordinal scale. Additionally, since there is little difference between the results of the mean and the median values, we unified the notation to the median.

Reviewer 3 Report
In this article, the authors seek to answer how carbapenem consumption in various categories of Japanese hospitals relates to a set of outcomes, a mix of «care variables» recorded at hospital discharge and a series of diagnosis/procedure groups. All data, both clinical/administrative and carbapenem measures, are extracted from the national diagnosis/procedure combination (DCP) payment system database from the year 2020. The number of hospitals included in this study is vast, with only about 8% of all Japanese hospitals excluded due to missing data.
I read with interest reference 32, «Development of the predicted and standardised carbapenem usage metric: Analysis of the Japanese Diagnosis Procedure Combination payment system data published in the J Infect Chemother (2020). In this short article, you found similar results as in the present paper with data from 2017, using fewer outcome variables and including a smaller number of hospitals.
I would first ask you to confirm that the metric you use for carbapenem consumption is DDDs per 1000 bed days occupied by in-patients and not DDDs per inhabitants/day of the catchment area, which seems to be a metric used in some studies.
Another central concept issue is the meaning of «DPC-prepared» versus «non-DPC» hospitals. You base all analyses of all hospitals on these DPC groups, so how is this possible if the largest number of hospitals are «non-DPC? What about the university and community hospitals? Are they not DPC-prepared?
I wish to comment on some other methodological issues, although you partly address them as limitations of the study in the Discussion (lines 191-196). My concern is the use of DPC payment system as a source for major independent variables used to establish determinants for carbapenem use. First, DPC is probably not well known among many international readers, few of which I presume will seek out the cited references. Therefore, I suggest you discard Figure 1, which I think adds little to the text, and in its place, seek to illustrate the logic behind the DPC system. Secondly, DPC seems to be a more elaborate variant of the US-invented DRG (Diagnosis Related Groups), also developed for fiscal purposes. As such, it encompasses all the principal ICD-10 diagnoses with no emphasis on infectious conditions, so I propose that the relation to broad-spectrum antibiotic use is somewhat dubious. However, the MDC groups might serve as a proxy for specialities where carbapenem is commonly used. You also discussed this issue in the 2020 article (ref. 32), and the same (entirely plausible) MDC’s you then found related to high carbapenem use were the same in the present article, namely respiratory and haematological/immunological conditions.
What puzzles me is that «non-DPC hospitals» has a whole series of MDC groups positively associated with carbapenem use, while the university hospitals have only 1-2 but markedly related groups. Can you elaborate on or substantiate your explanation in the Discussion (lines 166-168) that Uni hospitals have «similar medical functions and inpatient backgrounds»? I do not understand this. Are these institutions really so narrowly specialised?
For all non-university hospitals, a strong association between readmission at 14 days and carbapenem use was demonstrated by linear regression. Since your data does not include specific information on disease severity or the extent of patients' underlying diseases, it seems plausible that a high carbapenem use may be a marker for a high disease burden, which translates well into a pattern of frequent admissions. You refer to this as a «contradicting» result (in lines 154-158), as if carbapenem, through its clinical efficacy, would prevent readmissions – but it hardly cures underlying cancers. We probably agree on this, but I wish you could give a more precise statement.
In the Abstract, you state that «…a problem of resistant bacteria due to overuse was found» (lines 25-26). Unfortunately, I can not find any data or factual statements to this end in the article text other than speculations of overuse as deduced from a large number of outlier hospitals with excessive carbapenem use in Figure 2 (e.g., lines 86-87, 180-183). There is no reference to data on bacterial resistance in your text.
Some more specific concerns:
Table 1:
1) In the % of different outcomes at discharge, you list the two categories «Stable diseases» and «Others», which are not mentioned in the Methods section (lines 211-212).
2) What do the «Others» outcomes represent (a large group in most hospitals), and how do you define «Stable diseases»; is this a specific variable in the administrative database, or do you somehow base it on the principal diagnoses in the MDC categories?
3) Remission (in cancer treatment?) is a tiny percentage at discharge in all hospitals. Could this group be merged with «Stable diseases» for analytic purposes?
4) Again, for simplicity, I do not understand why you include the small number of licenced beds for long-term care and psychiatry in the analyses. (Are beds for psychiatry only located in university hospitals?) These patients are hardly using carbapenems, at least not in my experience.
5) In the linear regression, do you use «Cure»/»Not cure» as a binary variable, or are the «no-cure» variables treated as independent variables? Please clarify.
Figure 2:
It is not easy to understand which scatterplot relates to the numeric data in these cells, although «diagonal cells» in the figure text give a hint. The box text is tiny. Maybe the scatterplots could be omitted, and the data be shown in a table, not least because there were correlations.
Statistical analysis:
My concern for Table 1 # 5 above seems to be answered in lines 234-237, where you list the covariates chosen for the linear regression. However, you write «discharge with cure» twice (line 235). Would the correct text be «discharge without cure» for one of the entries?
Author Response
Reviewer #3:
Thank you very much for reviewing our manuscript together with the comments. The suggestions for improving the manuscript were very helpful in preparing a revised submission. We revised the manuscript according to your suggestions.
1. I would first ask you to confirm that the metric you use for carbapenem consumption is DDDs per 1000 bed days occupied by in-patients and not DDDs per inhabitants/day of the catchment area, which seems to be a metric used in some studies.
Response:
Thank you for the comment. DDDs per inhabitant/days are used to evaluate antibiotic use in the specific region. This study focused on only in-hospital carbapenem use. Therefore, we believe DDDs/bed days targeted for in-patients are suitable for analysis of associated factors for carbapenem use.
2. Another central concept issue is the meaning of «DPC-prepared» versus «non-DPC» hospitals. You base all analyses of all hospitals on these DPC groups, so how is this possible if the largest number of hospitals are «non-DPC? What about the university and community hospitals? Are they not DPC-prepared?
Response:
Thank you for the comment. Approved by the Ministry of Health, Labour and Welfare (MHLW) based on various criteria is required to approve as DPC-hospital. DPC-prepared hospitals have been prepared for application to DPC and the data was the distinction by the MHLW. All university hospitals are approved as DPC hospitals in Japan and applicate a specific reimbursement system. In addition, large-sized hospitals with equivalent to a university hospital that approved by the MHLW in the DPC community hospitals also applicate a specific reimbursement system.
3. I wish to comment on some other methodological issues, although you partly address them as limitations of the study in the Discussion (lines 191-196). My concern is the use of DPC payment system as a source for major independent variables used to establish determinants for carbapenem use. First, DPC is probably not well known among many international readers, few of which I presume will seek out the cited references. Therefore, I suggest you discard Figure 1, which I think adds little to the text, and in its place, seek to illustrate the logic behind the DPC system. Secondly, DPC seems to be a more elaborate variant of the US-invented DRG (Diagnosis Related Groups), also developed for fiscal purposes. As such, it encompasses all the principal ICD-10 diagnoses with no emphasis on infectious conditions, so I propose that the relation to broad-spectrum antibiotic use is somewhat dubious. However, the MDC groups might serve as a proxy for specialities where carbapenem is commonly used. You also discussed this issue in the 2020 article (ref. 32), and the same (entirely plausible) MDC’s you then found related to high carbapenem use were the same in the present article, namely respiratory and haematological/immunological conditions.
Response:
We agree with the reviewer's comment. The Japanese administrative database is not developed for analysis of infectious diseases and antimicrobial use. Therefore, we would like to propose the provide these specific data against policymakers and stakeholders based on this study results.
We added more explanation of DPC system in the introduction section.
Change: P2 L62‐68
Medical institution categories have been classified as the university hospital group, university-equivalent community hospital group, and community hospital group. Over 60% of hospitals in Japan submit “DPC data” to the Ministry of Health, Labour, and Welfare (MHLW). DPC data include discharge abstract and administrative claims data of inpatients. The MHLW collects the data for the purpose of health policy planning including the DPC-based reimbursement system.
4. What puzzles me is that «non-DPC hospitals» has a whole series of MDC groups positively associated with carbapenem use, while the university hospitals have only 1-2 but markedly related groups. Can you elaborate on or substantiate your explanation in the Discussion (lines 166-168) that Uni hospitals have «similar medical functions and inpatient backgrounds»? I do not understand this. Are these institutions really so narrowly specialised?
Response:
Thank you for the comment. As above mentioned, university-equivalent community hospitals are approved by the MHLW as having functions equivalent to those of a university hospital. Therefore, there is no difference in the function of the medical institution, and providing comprehensive medical care. These may not need to be analyzed separately, because they are the standards of the Japanese healthcare system. However, health policy is determined for each DPC category. So, we considered it is important for stratified by DPC category.
5. For all non-university hospitals, a strong association between readmission at 14 days and carbapenem use was demonstrated by linear regression. Since your data does not include specific information on disease severity or the extent of patients' underlying diseases, it seems plausible that a high carbapenem use may be a marker for a high disease burden, which translates well into a pattern of frequent admissions. You refer to this as a «contradicting» result (in lines 154-158), as if carbapenem, through its clinical efficacy, would prevent readmissions – but it hardly cures underlying cancers. We probably agree on this, but I wish you could give a more precise statement.
Response:
We agree with the reviewer's comment. We think that the increment of readmission would be caused by disease severity and complicated comorbidities such as terminal cancer etc. We revised "contradicting" to "inconsistent".
Change: P6 L148
6. In the Abstract, you state that «…a problem of resistant bacteria due to overuse was found» (lines 25-26). Unfortunately, I can not find any data or factual statements to this end in the article text other than speculations of overuse as deduced from a large number of outlier hospitals with excessive carbapenem use in Figure 2 (e.g., lines 86-87, 180-183). There is no reference to data on bacterial resistance in your text.
Response:
Thank you for the comment. We deleted the sentence.
Change: P1 L25‐26
Specific concerns
Table 1:
1) In the % of different outcomes at discharge, you list the two categories «Stable diseases» and «Others», which are not mentioned in the Methods section (lines 211-212).
Response:
Thank you for the comment. We revised it.
Change: P7 L206‐207
2) What do the «Others» outcomes represent (a large group in most hospitals), and how do you define «Stable diseases»; is this a specific variable in the administrative database, or do you somehow base it on the principal diagnoses in the MDC categories?
Response:
These outcomes have been defined by DPC database. "Stable diseases" defined as "treatment for the disease has been performed, but no improvement has been seen and the disease is considered to be unchanging" by the MHLW. These outcome diagnose by attending physician.
3) Remission (in cancer treatment?) is a tiny percentage at discharge in all hospitals. Could this group be merged with «Stable diseases» for analytic purposes?
Response:
Integration is possible, but the result will be consistent. We would like to analyze the data according to the classification of the administrative database.
4) Again, for simplicity, I do not understand why you include the small number of licenced beds for long-term care and psychiatry in the analyses. (Are beds for psychiatry only located in university hospitals?) These patients are hardly using carbapenems, at least not in my experience.
Response:
We agree with reviewer's comment. Reviewer #2 pointed out similarly. Although we collected the data from the DPC database, these factors are not considered relevant to carbapenem consumption typically. Moreover, almost all of the hospitals in this study do have not these beds as shown in Table. We excluded the factors of long-term care beds and psychiatric beds. We performed a regression analysis again and the results were identical. The descriptions related to these have been revised.
Change: Table 1 and P8 L234 -235
5) In the linear regression, do you use «Cure»/»Not cure» as a binary variable, or are the «no-cure» variables treated as independent variables? Please clarify.
Response:
"Cure" is cure rate in each hospital as continuous variable. We added explanation.
Change: P7 L206‐207 and Table 3
Figure 2:
It is not easy to understand which scatterplot relates to the numeric data in these cells, although «diagonal cells» in the figure text give a hint. The box text is tiny. Maybe the scatterplots could be omitted, and the data be shown in a table, not least because there were correlations.
Response:
Thank you for the comment. We replaced the figure 3 with table 2.
Statistical analysis:
My concern for Table 1 # 5 above seems to be answered in lines 234-237, where you list the covariates chosen for the linear regression. However, you write «discharge with cure» twice (line 235). Would the correct text be «discharge without cure» for one of the entries?
Response:
Thank you for pointing that out. We apologize for the mistake. We corrected the duplicated "discharge with cure".
Change: P8 L232

Round 2
Reviewer 2 Report
Thank you for considering my comments! I have some minor suggestions:
- L107-108 “No strong correlation was found between 107 DDDs/patient days and each clinical outcome in each facility type” you should clarify the cut-off and change your sentence as “No strong correlation was found between 107 DDDs/patient days and each clinical outcome in each facility type (all rho<0.5)”
- In Table 3, you could add one subtitle for each different model
|
Factors |
Partial regression coefficient (95% confidence interval) |
p-value |
|
Model 1 (University hospitals) |
||
|
… |
|
|
|
Model 2 (University hospital-equivalent community hospitals) |
||
|
… |
|
|
- L221 “multivariable liner regression” you should correct it to “multivariable linear regression”
- L214 “Hospital characteristics of each facility type were presented as median and interquartile range.” I agree with median, you should report that you checked for normality i.e. “Hospital characteristics of each facility type were presented as median and interquartile range, due to non-normal distribution”
Author Response
Reviewer #2:
Thank you very much for reviewing our manuscript together with the comments. The suggestions for improving the manuscript were very helpful in preparing a revised submission. We revised the manuscript according to your suggestions.
- - L107-108 “No strong correlation was found between 107 DDDs/patient days and each clinical outcome in each facility type” you should clarify the cut-off and change your sentence as “No strong correlation was found between 107 DDDs/patient days and each clinical outcome in each facility type (all rho<0.5)”
Response:
Thank you for the comment. We added the information.
Change: P4 L108
- - In Table 3, you could add one subtitle for each different model
|
Factors |
Partial regression coefficient (95% confidence interval) |
p-value |
|
Model 1 (University hospitals) |
||
|
… |
|
|
|
Model 2 (University hospital-equivalent community hospitals) |
||
|
… |
|
|
Response:
Thank you for the comment. We have revised it.
Change: Table 3
- - L221 “multivariable liner regression” you should correct it to “multivariable linear regression”
Response:
Thank you for pointing that out. We apologize for the mistake. We corrected the "multivariable linear regression".
Change: P7 L221
4. - L214 “Hospital characteristics of each facility type were presented as
median and interquartile range.” I agree with median, you should report that you checked for normality i.e. “Hospital characteristics of each facility type were presented as median and interquartile range, due to non- normal distribution”
Response:
Thank you for the comment. We added the information.
Change: P7 L215
Reviewer 3 Report
No further comments, nice work.
Author Response
Thank you very much for reviewing our manuscript together with the comments.
The suggestions for improving the manuscript were very helpful in preparing a revised submission.